# Guide to Metabolomics Analysis: A Bioinformatics Workflow

**DOI:** 10.3390/metabo12040357

**Published:** 2022-04-15

**Authors:** Yang Chen, En-Min Li, Li-Yan Xu

**Affiliations:** 1The Key Laboratory of Molecular Biology for High Cancer Incidence Coastal Chaoshan Area, Shantou University Medical College, Shantou 515041, China; cy_koasde@163.com; 2Department of Biochemistry and Molecular Biology, Shantou University Medical College, Shantou 515041, China; 3Guangdong Provincial Key Laboratory of Infectious Diseases and Molecular Immunopathology, Institute of Oncologic Pathology, Shantou University Medical College, Shantou 515041, China

**Keywords:** metabolomics, metabolomics analysis tools, metabolic pathways summary, multi-omics integration algorithms

## Abstract

Metabolomics is an emerging field that quantifies numerous metabolites systematically. The key purpose of metabolomics is to identify the metabolites corresponding to each biological phenotype, and then provide an analysis of the mechanisms involved. Although metabolomics is important to understand the involved biological phenomena, the approach’s ability to obtain an exhaustive description of the processes is limited. Thus, an analysis-integrated metabolomics, transcriptomics, proteomics, and other omics approach is recommended. Such integration of different omics data requires specialized statistical and bioinformatics software. This review focuses on the steps involved in metabolomics research and summarizes several main tools for metabolomics analyses. We also outline the most abnormal metabolic pathways in several cancers and diseases, and discuss the importance of multi-omics integration algorithms. Overall, our goal is to summarize the current metabolomics analysis workflow and its main analysis software to provide useful insights for researchers to establish a preferable pipeline of metabolomics or multi-omics analysis.

## 1. Introduction

Metabolomics is a rapidly evolving field that deals with the high-throughput characterization of metabolites, and is the study of the metabolite composition of cell types, tissues, organs, or organisms [1,2]. Metabolomics is the collection of endogenous small molecules that mark specific fingerprints of cellular biochemistry [3]. It measures numerous low-molecular weight metabolites, such as amino acids, sugars, fatty acids, lipids, and steroids [4]. Small modifications in the chemical structure and some external stimuli (e.g., infections and allergens) can dramatically change the function of a metabolite [5,6,7]. Metabolites, in addition to being produced directly by the host organism, can be derived by host microbiota or transformed from dietary, xenobiotic or other exogenous sources [8]. It is worth emphasizing that lipids are important metabolites in the organism, and have a wide variety of properties, such as insolubility in water and solubility in non-polar organic solvents [9]. Lipids are involved in the regulation of many physiological reactions. An important step in energy metabolism is the hydrolysis of triglycerides (TG) in lipid droplets to release fatty acids that can be used or stored [10]. Abnormal lipid metabolism can lead to many diseases, such as obesity, atherosclerosis and diabetes [11,12,13]. The concept of “lipidomics” was first introduced by Richard et al. in 2003, and was followed by the introduction of more efficient and accurate lipidomics research [14]. In 2005, Markus Wenk also suggested that more and more studies show a direct link between lipids and many diseases, and that metabolomics and proteomics studies are no longer sufficient to provide a clear picture of the causes of these diseases [15]. Therefore, it has become important to link lipid studies to diseases, with lipidomics no longer being just a single study of lipids, but rather a comprehensive discipline linking the proteome, metabolome, and various diseases. Lipidomics has been used to elucidate the metabolism of lipids by studying their composition, structure, and quantification in biological samples, and can be used to search for biomarkers and to study the mechanisms of lipids in various phenomena [10,16]. With the continuous innovation of mass spectrometry technology and the increasing awareness of the importance of the biological functions of lipids, lipidomics, as an important branch of metabolomics, has gradually attracted the attention of researchers [17].

Metabolomics is wildly used in cancers and metabolism-related diseases. For example, researchers found that, in bladder cancer, the metabolites in the tricarboxylic acid (TCA) cycle were significantly changed [18,19,20,21,22], along with changes in fatty acid metabolism [19,20,21,23]. In colorectal cancer, disordered methionine metabolism and abnormal TCA cycle function have been reported [24]. Amino acid metabolism [25,26,27,28,29,30], bile acid metabolism [25,31], choline metabolism [25,26,27,28], fatty acid metabolism [27,28,29,30], and glycolysis [32,33] have also been found to be abnormal in liver cancer through metabolomics analysis. In diabetes, many metabolic pathways have been found to be disordered, such as acetoacetate metabolism [34], acylcarnitine metabolism [35,36,37], palmitic acid metabolism [38,39,40], linolenic acid metabolism [38], cholesterol metabolism [41], carbohydrate metabolism [34,42,43,44], glycine and serine metabolism [45,46], and fatty acid metabolism [40,44,47,48,49]. Glycolysis [50,51], the TCA cycle [47,52,53,54], the urea cycle [55], and glutathione metabolism [56] have also been found to be abnormal in obesity. In Alzheimer’s disease, abnormal amino acid metabolism [57,58,59,60,61], fatty acid metabolism [57], linoleic acid metabolism [62], cholesterol metabolism [57], glycine and serine metabolism [63], aspartate metabolism [64], glycerophospholipid metabolism [65] and polyamine metabolism [57] have been reported. In this context, metabolomics is useful in the identification of biomarkers associated with the diagnosis/prognosis of different oncological processes and the response to treatment [66,67].

Previous reviews have summarized metabolomic analysis platforms [68], and some have summarized metabolomic data preprocessing methods [69]. However, a detailed guide for the process of metabolomics analysis would be an essential addition to these previous works. This review aims to describe the overall metabolomics analysis process and summarize the currently available software and databases for analyzing metabolomics data, thus providing a standard protocol for analyzing metabolomics data to identify clinically or disease-relevant biomarkers for researchers.

## 2. The Analysis Workflow of Metabolomics

The specific characteristics of metabolomics data require the application of different bioinformatics tools following a specific workflow (Figure 1). The first step in the metabolomics workflow involves using different techniques to isolate and characterize different groups of metabolites. There are two main platforms of metabolomics analysis: mass spectrometry (MS) and nuclear magnetic resonance (NMR) spectroscopy. Each has its own advantages and disadvantages [70]. MS-based metabolomics is generally preceded by a separation step, which reduces the complexity of the biological sample and allows MS analysis of different sets of molecules at different time [71]. MS acquires spectral data in the form of a mass-to-charge ratio (*m*/*z*) and a relative intensity of the ionized compound [72]. The most common separation techniques in MS technology are liquid chromatography (LC) and gas chromatography (GC) columns (LC-MS and GC-MS, respectively) [73]. MS allows for the reliable identification of metabolites. Especially when MS is used in tandem with chromatographic separation methods, its resolving ability is improved. At the same time, MS has a short analysis cycle (with an analysis time ranging from 5 to 140 min) and allows for selective qualitative and quantitative analyses. Therefore, this technique is the most widely used. The main disadvantages of MS are the high cost of the instruments and the requirement for sample separation or purification prior to putting the sample into the mass spectrometer. NMR is a spectroscopic technique based on the principle of energy absorption and re-emission by atomic nuclei due to variations in an external magnetic field [74]. The spectral data generated by NMR can be used to quantify the concentration and characterize the chemical structure of metabolites. The advantages of NMR are that it is a nondestructive and highly reproducible technique, and does not require extensive sample preparation [75]. In particular, NMR has the ability to provide a high degree of structural information in a short time. Nevertheless, NMR has a lower sensitivity, which means that lower concentrations of potentially important compounds can be masked by larger peaks, and thus cannot be identified [76]. The identification and quantification of metabolites is a complex task that cannot be easily automated. Therefore, careful data processing and statistical analysis are required to derive useful and reliable information from these profiles [77]. The application of NMR spectroscopy is not limited to liquid and solid samples but extends to intact tissue samples with high-resolution magic angle rotation (HRMAS) NMR spectroscopy [78,79]. LC-MS is more suitable for the detection of moderately polar compounds and substances with high polarity, in terms of specific types of substances, such as fatty acids, alcohols, phenols, vitamins, organic acids, polyamines, nucleotides, polyphenols, terpenes, flavonoids, lipids, and other compounds [73,80]. The inherent limitation of GC-MS is that it only detects volatile compounds or compounds that can be derivatized into volatiles. GC-MS can detect amino acids, organic acids, fatty acids, sugars, polyols, amines, sugar phosphates, and other substances [81,82]. It is worth noting that GC-MS measurement of water-soluble substances requires derivatization because GC-MS analysis can only be performed directly if the sample is volatile and stable to heat. Each separation method has its own resolution and sensitivity in the identification of metabolites, and its selection is based on the chemical and physical properties of each sample or the chemical and physical characteristics of the hypothetical target compounds, as well as the type of analysis to be performed (untargeted or targeted) [83,84].

The second step is the preprocessing of raw signals (chromatograms, spectra, or NMR data) by specific software for quantitative analysis of compounds. (e.g., XCMS [85], MAVEN [86] or MZmine3 [87]). In general, this step includes noise reduction, retention time correction, peak detection and integration, and chromatographic alignment. Several main platforms of the steps are discussed in Section 4.

In the third step, data processing is performed, and quality control (QC) is necessary. Data from QC samples are used to separate different-quality (high or low) data, balance the analytical platform’s bias, and correct for noise in the signal. QC samples are used to determine the variance of metabolite features. If the variance of a feature is too high, it will be removed from the analysis [69,73]. Then, data normalization is used to reduce systematic bias or technical variation and to avoid misidentification due to disparate input of large amounts of metabolomics data. Subsequently, mass spectrometry peak data are used for compound identification by comparing it to authentic standard data (typically through an in-house library). In the absence of an in-house library, researchers can also apply public databases for compound identification. It is worth mentioning that researchers should be aware of the criteria for reporting metabolite annotation and identification. The Metabolomics Standards Initiative (MSI) was conceived in 2005 with the aim of enabling the effective application, sharing and reuse of data [88]. The standard proposes four different levels of metabolite identification observed in the scientific literature, including identified metabolites (level 1), presumptively annotated compounds (level 2), presumptively characterized compound classes (level 3), and unknown compounds (level 4). It is recommended that researchers define identification levels, common names, and structure codes (e.g., InChI or SMILES) in their publications and when submitting data to the repository [88]. For untargeted metabolomics studies, different databases, such as the Human Metabolome Database (HMDB [89]) or the Metabolite and Tandem MS Database (METLIN [90]), are used to identify metabolites from spectra.

Typical statistical analyses of metabolomics data include univariate and multivariate approaches. They enable the evaluation of the input metabolomic dataset and identification of metabolites that undergo abnormal changes. Subsequently, functionally relevant metabolites can be distinguished by further data mining methods [91]. Traditional statistical methods determine the relationships between variables based solely on mathematical criteria, without fully taking into account biological correlations, which is one of their limitations [92]. So, the combined use of multiple statistical techniques is recommended when performing metabolomics analysis. In such a context, an appropriate *p*-value is used to rank the significantly expressed metabolites and determine a reliable threshold to select the most significant one. The choice of this threshold may influence the final biological interpretation, and is therefore particularly critical [93].

In the next step, the screened metabolites are linked to their biological context by pathway and enrichment analysis. The aim of enrichment analysis is to explore the profile of functionally relevant metabolites to determine the link between changes in metabolite expression and biological context. This allows the use of a list of altered metabolites to suggest biological pathways or disease conditions that would indicate the subsequent steps in the study. The goal of pathway analysis is to identify pathways that have a significant impact on a specific biological process. Enrichment and pathway analyses are performed using specialized software tools [94] that map metabolites to known biochemical pathways based on information in public databases such as KEGG [95]. Subsequently, investigators typically use network visualization tools to present and understand their results.

Multi-omics data integration and analysis pipelines for studying the pathogenesis of disease and the influence of environmental risk factors are scarce. In this last step, an integrated multi-omics platform provides a reliable and understandable overview of metabolic changes [94]. The identified metabolites and metabolic pathways can be integrated with other omics data, which may help us to obtain more comprehensive information about the biological phenomena.

## 3. Statistical Analysis in Metabolomics

Depending on the experimental context, various types of data mining and statistical methods can be applied to metabolomics data. In the following overview, we summarize in detail the univariate and multivariate statistical analysis methods applied in metabolomics.

### 3.1. Univariate Analysis

Univariate analysis usually provides a preliminary overview of data characteristics that may be important in identifying the conditions under study. For two-group data (both unpaired and paired analyses), we can perform fold change analysis, *t*-tests, and volcano plots. For multi-group data, we can perform one-way analysis of variance (ANOVA), as well as related post hoc analysis and correlation analysis. Since each patient or biological sample usually has a large number of metabolites, and each metabolite needs to undergo a separate statistical test, a large number of false positive results can be obtained through multiple tests. To reduce this, multiple testing methods (e.g., Bonferroni, Bonferroni–Holm and Benjamini–Hochberg corrections) must be used to correct for *p*-values [96]. The Benjamin–Hochberg correction, also known as the false discovery rate (FDR), is one of the recommended methods because it allows for controlling the proportion of false positives in all significant results.

### 3.2. Multivariate Analysis

Since multi-omics data usually contain some characteristics that vary with phenotype or experimental conditions, the use of multivariate analyses that allow simultaneous observation and analysis of more than two statistical variables is recommended. Multivariate analysis includes multiple variance analysis (ANOVA), multiple regression analysis, factor analysis, principal component analysis (PCA), partial least squares discriminant analysis (PLS), cluster analysis, and machine learning (e.g., random forest and SVM). Because multivariate analysis uses the weighted averages to summarize the original variables in fewer variables, they are useful for exploratory data analysis. PCA analysis starts from the interrelationship between the original variables, linearly transforms them to several independent composite indicators (i.e., principal components) according to the principle of variance maximization, takes two to three principal components for graphing, visually describes the differences in metabolic patterns and clustering results between different groups, and searches for the original variables that contribute to intergroup classification as biomarkers through loadings plots. PCA is commonly used as a pre-analysis and quality control step for metabolomics data to observe whether there are intergroup classification trends and data outlier points. PCA can also be used to analyze whether quality control samples are clustered together, or if they are scattered or have some variability, which would indicate problems with the quality of the assay. For example, Pasikanti et al. used PCA to analyze urine bladder cancer metabolomics data and observed that the QC samples were tightly clustered on the PCA score plot, thus validating the stability of the instrument’s assay and the reliability of the metabolomics data [19]. PLS-DA is another commonly used classification method in metabolomics data analysis, which combines regression models with dimensionality reduction and discriminant analysis of regression results using certain discriminant thresholds. The difference between this dimensionality reduction method and PCA is that PLS-DA decomposes both the independent variable X matrix and the response variable Y matrix, and uses its covariance information in the decomposition so that the dimensionality reduction effect can extract the inter-group variation information more efficiently than PCA. In practice, a PLS-DA score plot is often used to visualize the classification effect of the model, and the greater the separation of the two groups in the plot, the more significant the classification effect [97]. It is important to note that cross-validation should be performed when using PLS-DA, and the PLS-DA results should be interpreted together with PCA to avoid overfitting problems in the metabolomic data. The goal of the metabolomic analysis is to screen potential biologically relevant markers to explore the metabolic mechanisms involved, and therefore requires variable screening with the help of certain feature screening methods. Random forest (RF) and SVM provide very flexible models for handling data with many covariates [97]. For example, RF is a nonparametric ensemble approach that prioritizes predictions by trying to find nonlinear patterns in metabolites that can explain the variation in each outcome [97]. RFs are very powerful tools if the relationship between metabolites and outcomes is complex and nonlinear, and have been used for missing data interpolation and outcome analysis in metabolomics [98,99]. A disadvantage of this approach is that as with PCA, it does not provide a measure of statistical significance or provide any *p*-values or equivalence measures. Nevertheless, RFs can provide analysts with a ranked list of the most important metabolites. All the mentioned machine learning methods have many of the same limitations as RF, as they can provide variable importance measures, but not a set of variables that can be considered statistically significant [97]. To evaluate the importance of each variable more objectively and comprehensively, a combination of the above methods is generally adopted in metabolomics studies for variable screening. A more common strategy is to perform univariate analysis, then combine the variable importance scores from multivariate models as screening criteria, and finally integrate them with the variables screened by machine learning models, such as selecting variables with FDR ≤ 0.05 and VIP > 1.5, and that are ranked high in the RF as potential biomarkers.

## 4. Software Tools for Metabolomics Data Analysis and Integration

Metabolomics analyses need powerful software tools to address the vast amount and variety of data. Excellent metabolomics software should include one or more of the following functions: (1) the ability to process of raw spectral data, (2) statistical analysis to find significantly expressed metabolites, (3) the ability to connect to metabolite databases for metabolite identification, (4) bioinformatics analysis and visualization of molecular interaction networks, and (5) the ability to integrate and analyze multi-omics data. We searched the PubMed literature database with the following keywords: “metabolomics software”, “metabolomics & bioinformatics”, “metabolomics analysis tool”, “lipidomic software”, “metabolomics protocol”, “multi-omics software”, and “multi-omics algorithm”. Only original research articles regarding software and databases were included. In this section, we introduce several data analysis tools (Table 1) and show their characteristics and compare their advantages and disadvantages (Figure 2 and Figure 3).

### 4.1. MS-DIAL

MS-DIAL was previously developed as free data pre-processing software for LC-MS data processing, but now the MS-DIAL 4.0 tool can also process LC-MS, GC-MS, and NMR data, in particular to obtain deconvoluted spectra from high-resolution GC-MS data as a prerequisite for compound identification (MS-DIAL 4.0, Hiroshi Tsugawa, Kanagawa, Japan) [100,101]. MS-DIAL offers multiple data-acquisition processing and includes the spectra for compound ID. It also includes normalization and statistical analysis options (http://prime.psc.riken.jp/, accessed on 31 March 2022) [100]. MS-DIAL has an internal GC/MS database, as well as silica retention time and MS/MS database for LC-MS/MS-based lipidomics [101].

### 4.2. MZmine 3

MZmine 3 (Tomáš Pluskal, Prague, Czech) is an open source software for mass spectrometry data processing that focuses on LC-MS data but can still handle GC-MS and NMR data (http://mzmine.github.io/, accessed on 13 January 2022). This software includes a complete workflow for LC-MS data analysis, including raw data processing, data filtering and peak identification, isotope detection, statistical analysis, and visualization [87].

### 4.3. El-MAVEN

El-MAVEN (Shubhra Agrawal, Cambridge, USA) is an open source desktop software for processing LC-MS, GC-MS and NMR data labeled in open formats (mzXML, mzML, CDF) [102]. This software has a graphical and command line interface, integrates with a cloud-based platform for storage, and conducts further analyses, such as relative fluxes and quantification [102]. El-MAVEN features a multi-file chromatography comparator, a peak feature detector, and an isotope calculator. El-MAVEN is more powerful, faster, and more user-friendly than Maven, and includes an additive calculator, fragment spectra matching, and peak editor. The El-MAVEN installer is available for Windows and Mac OS (www.elucidata.io/el-maven, accessed on 31 March 2022). Users can download the latest versions of these platforms-. Additionally, developers can follow the instructions to build El-MAVEN on Windows, Ubuntu, or Mac OS to set up the development environment (64-bit platforms only).

### 4.4. LipidMatch

LipidMatch can be used to annotate lipids detected by LC-MS (http://secim.ufl.edu/secim-tools/lipidmatch/, accessed on 31 March 2022). The LipidMatch fragment library contains over 250,000 lipid species spanning over 50 lipid types [103]. Users can annotate lipids in feature tables generated by its optimized peak picking and filtering strategy. LipidMatch is also used for the annotation of direct infusion and imaging experiments. The software is modular, which makes it suitable for a variety of workflows, and researchers can use it with a variety of peak picking software (e.g., MZmine 3, XCMS (Gary Siuzdak, California, USA), and MS-DIAL 4.0). LipidMatch also provides its lipid libraries in csv format and the R scripts for LipidMatch.

### 4.5. LipiDex

LipiDex (Joshua J Coon, Madison, WI, USA) is a unified software that can be used for lipid identification by LC-MS/MS. It has the ability to greatly reduce manual processing bias and improve the confidence of identification [104]. When using LipiDex, researchers first create a library of lipid spectra, then use fragment templates to build composite lipid spectra and mass spectrometry fragment models, and subsequently correlate spectral identifications with chromatographic peaks to generate LC-MS/MS lipidomic datasets with high confidence. LipiDex can automatically filter peak lists for additive peaks, endogenous fragments, and dimers (https://github.com/coongroup/LipiDex, accessed on 31 March 2022).

### 4.6. MetFlow

MetFlow is a web-based tool developed in 2019 (http://metflow.zhulab.cn/, accessed on 13 January 2022) [105]. It offers a standardized workflow for metabolomics data processing and is an interactive web server. Researchers can also use it to perform data cleaning and differential analysis. Its functions include: (1) batch alignment, (2) data quality check and visualization, (3) missing value processing and outlier removal, (4) data normalization and integration, (5) statistical analysis, (6) performance validation, and (7) pathway enrichment analysis. The software enables users with little knowledge in programming and statistics to perform metabolomics data analysis. MetFlow is simple to operate. It has excellent graphic visualization ability (Figure 2a–f) and it can verify the results by uploading test data. However, its disadvantages are that the uploaded file format is fixed, and its pathway enrichment analysis cannot provide the visualization of specific pathways. Therefore, we cannot intuitively find the role of metabolites in the pathway.

### 4.7. MetaboAnalyst 5.0

MetaboAnalyst 5.0 is a comprehensive, freely accessible web-based metabolomics analysis platform (https://www.metaboanalyst.ca/, accessed on 13 January 2022). It was first developed in 2009 [106], then updated in 2012 (MetaboAnalyst 2.0 [107]), in 2015 (MetaboAnalyst 3.0 [108]), in 2019 (MetaboAnalyst 4.0 [94]) and in 2021 (MetaboAnalyst 5.0 [109]). It can be locally installed at the same time. MetaboAnalyst provides comprehensive online tools for metabolomics data analysis, statistical analysis, functional annotation, and visualization of data. MetaboAnalyst 5.0 improves its analytical performance and user interactivity. The platform provides four major functional modules that can be classified into 12 categories: (1) statistical analysis (statistics, biomarker analysis, multifactor/time series analysis, power analysis); (2) functional analysis (metabolome enrichment analysis, metabolic pathway analysis, mass spectrometry peak prediction of pathway activity); (3) data integration and systems biology (biomarker meta-analysis, joint-pathway analysis, and network explorer) and (4) data processing and utility functions (compound ID conversion, batch effect correction, lipidomics, and links to several spectra analysis tools). The advantages of MetaboAnalyst 5.0 are that it supports several formats of uploaded data, and the statistical methods are more selective (Figure 2g–i). The wide variety of pathway analysis methods can also meet most needs (Figure 2j–l). MetaboAnalyst 5.0 has a corresponding R package, which greatly improves the autonomy of metabolomics analysis. In addition, multiple databases are linked for multi-omics analysis. Nevertheless, MetaboAnalyst 5.0 did not have the analysis module for integration of the metabolome and microbiome, which is a disadvantage of most metabolomics analysis software.

### 4.8. LipidSig

LipidSig is a web-based platform for the comprehensive analysis of lipidomic data [110]. It contains five main functions: (1) profiling (for pre-processing data), (2) differential expression, (3) machine learning, (4) correlation analysis, and (5) network. LipidSig can also create interactive plots and generate downloadable images and corresponding tables (http://chenglab.cmu.edu.tw/lipidsig/, accessed on 31 March 2022).

### 4.9. LION

LION/web enables statistical analysis of lipids. Additionally, the most powerful feature of the software is the integration of more than 50,000 lipids with biophysical, biochemical and cell biological features, allowing a comprehensive enrichment of lipids [111]. Additionally, the authors developed a web-based interface based on LION for easy operation by researchers (www.lipidontology.com, accessed on 31 March 2022).

### 4.10. METLIN

The METLIN tandem mass spectrometry (MS/MS) database was created in 2003 and made publicly available in 2005 [112] to help identify metabolites. At that time, no such database existed for identifying metabolites. In 2018, to improve the coverage of metabolites and help annotate them, in silico MS/MS spectra were generated on additional molecules in METLIN. These data were based on machine learning algorithms, the METLIN database, and the unique fragmentation information (provided by stable isotopes) [90]. METLIN is a free cloud-based platform and metabolite database. It has since grown from a small collection of MS/MS spectra on 100 metabolites in its first iteration to more than 10,000 metabolites in 2012 [113], with an additional 12,000 metabolites and compounds having been analyzed in the last 5 years. METLIN data are widely used in a variety of tandem mass spectrometry instrument types (https://metlin.scripps.edu/, accessed on 13 January 2022).

### 4.11. PaintOmics 3

PaintOmics 3 is a web-based resource for the integrated visualization of multi-omics data types on KEGG pathway diagrams (www.paintomics.org, accessed on 13 January 2022) [114]. PaintOmics 3 combines data analysis with data visualization, providing researchers with an efficient framework for their multi-omics data. Unlike other visualization tools, PaintOmics 3 covers a comprehensive pathway analysis workflow (Figure 3a,b), including automatic feature name conversion, multi-layered feature matching, pathway enrichment, network analysis, heatmaps, trend charts, and more. It accepts a wide variety of omics types, including transcriptomics, proteomics, and metabolomics, as well as region-based approaches such as ATAC-seq or ChIP-seq data. However, the input data need to be pre-processed.

### 4.12. 3Omics

3Omics is a web-based tool that was developed in 2013 (http://3omics.cmdm.tw, accessed on 13 January 2022). It is used to analyze, integrate, and visualize transcriptome, proteome, and metabolome human data [115]. 3Omics supports correlation analysis, phenotype mapping, pathway enrichment analysis, and co-expression analysis (Figure 3c). In fact, depending on the input data, the software offers four parts of integrated analyses: (1) transcriptomics, proteomics, and metabolomics (T-P-M), (2) transcriptomics and proteomics (T-P), (3) proteomics and metabolomics (P-M) and (4) transcriptomics and metabolomics (T-M). A single omics analysis mode is also available in the tool. 3Omics can also carry out text mining of the biomedical literature through information Hyperlinked Over Protein (iHOP [116]) to supplement missing information. The drawback is that pathway enrichment analysis cannot provide the visualization of specific pathways.

### 4.13. IMPaLa

IMPaLA is a web tool for transcriptomics, proteomics, and metabolomics pathway analysis (http://impala.molgen.mpg.de, accessed on 13 January 2022) [117]. It was developed in 2011. The web tool uses over 3000 pre-annotated approaches from 11 databases to perform over-expression or enrichment analysis on uploaded metabolites and gene lists. Therefore, it is possible to identify pathways that may be regulated at the transcriptional level, metabolic level, or both. The output results of the tool include a ranked list of pathways, the size of each pathway and the *p*-value and *q*-values from the joint analysis of genes and metabolites. By clicking on the pathway name, users will be guided to a summary web page at the source database. Results can also be downloaded as a tab-delimited file.

### 4.14. MetPA

MetPA is a user friendly, web-based tool for the analysis and visualization of metabolomics data (http://metpa.metabolomics.ca, accessed on 13 January 2022) [118]. It combines pathway enrichment analysis programs and pathway topology feature analysis to help identify the most relevant metabolic pathways (Figure 3d). The results are displayed in an interactive network visualization system that can be selected, dragged, and zoomed in and out. In addition, this tool offers a comprehensive compound library for metabolite name conversion, and it can also implement various univariate analyses. MetPA currently supports the analysis and visualization of 874 metabolic pathways in 11 common model organisms and it has been integrated into the MetaboAnalyst 5.0 platform.

### 4.15. MassTRIX

MassTRIX is a web-based software for metabolomics pathway enrichment analysis [119]. The input data of this tool require a mass peak list from high-precision MS experiments. MassTRIX marks the identified chemical compounds as differentially colored objects on the KEGG pathway maps (Figure 3e). Therefore, users can interpret the metabolic state of the organism based on the original organism and the true enzymatic capabilities in the case of submitted transcriptomics data. The tools’ output page summarizes the number of identified metabolites on all available pathways and gives a list of all metabolites that are annotated on any given pathway of the organism. Here, users should note that in some cases multiple alternative annotations may be found. The MassTRIX web server is freely accessible at http://masstrix.org (accessed on 13 January 2022).

### 4.16. MetaCore™

MetaCore™ (http://thomsonreuters.com/metacore/, accessed on 13 January 2022) is a commercial tool used as a web-based application. The software can analyze different kinds of high-throughput molecular data. MetaCore™ is also a high-quality database of mammalian biology, with collections including metabolites and other molecular classes, bioactive molecules and their interactions, signal transduction and metabolic pathways. It also enables genomic analysis, identifies potentially important variants, and provides data visualization, analysis, and data mining. Unfortunately, no detailed information is available on how MetaCore™ works. Therefore, our review of this tool is limited.

### 4.17. OmicsNet

OmicsNet (www.omicsnet.ca, accessed on 31 March 2022) can integrate different omics data based on molecular interaction knowledge and visualization using network analysis. It also can annotate SNPs, microbial taxa, or LC-MS peaks for network analysis [120]. The network analysis can contain genes, proteins, transcription factors (TF), miRNAs and metabolites, and the creation of different types of biological networks is derived from multiple molecular interaction databases (PPI, TF-gene, miRNA-gene, and metabolic protein interactions).

In general, MS-DIAL, Mzmine3, and EI-MAVEN can perform data preprocessing, normalization, identification, and statistical analysis of metabolomic data. MetFlow and MetaboAnalyst 5.0 can perform most of the metabolomics analyses, including data processing, statistical analysis, and pathway analysis. Other software platforms (OmicsNet, PaintOmics3, 3Omics, IMPaLA, MetPA, MassTRIX, and others) have performed well in the subsequent analysis, including multi-omics integration and pathway analysis. In lipidomics, LipidMatch and LipiDex can be used for lipid identification, and LipidSig can perform most of the lipidomic analyses and may be the better choice. After that, the LION/web could be used for enrichment analysis. Researchers can choose different analysis software according to their needs.

## 5. The Integration Algorithm of Multi-Omics Data

Due to the complex and multi-factorial context of metabolic diseases, the results of metabolomic analysis should be followed by novel techniques to link the overall effectiveness between organisms, metabolites, microbiota, and individuals. Currently, the different molecular levels can be systematically divided into genomics, transcriptomics, proteomics, metabolomics, and microbiology. Genomics allows the evaluation of the whole genome of an organism and the analysis of the localization and function of genes. Transcriptomics measures the expression of genes at a given time point. Subsequently, gene translation enables protein expression, giving rise to proteomics. Proteins can translate biologically active compounds (metabolites) into other metabolic molecules. Thus, metabolomics is an assessment of the level of metabolism in an organism, a process complicated by the fact that metabolites are in dynamic equilibrium and respond to external and internal factors. Finally, microbiomes characterize the gut microbial community [121]. Combining all of these analyses in the same biological context allows us to outline the interactions between multiple metabolite networks and gene expression markers in multiple tissues or locations, as well as determine the possible impact of microbial members on biosynthesis. Therefore, the development of an efficient and practical multi-omics algorithm is important to interpret the results of metabolomics.

In 2018, Pedersen et al. [122] proposed a calculation protocol, detailing and discussing dimensionality reduction technology and the subsequent method of integrating and interpreting multi-omics data. Dimensionality reduction of the different omics data was achieved through data normalization, the combination of co-abundant genes and metabolites, and the integration of existing biological knowledge. Using prior knowledge to overcome the functional redundancy among microbiome species is a major advancement of the method compared with existing alternative methods. Researchers can integrate multi-omics data with host physiology variables or any other phenotypes of interest to perform a three-pronged analysis to identify potential mechanistic connections through this framework and then test it through experimentation. Although it is a framework for a human metabolome-microbiome study, it is generalizable to other organisms and environmental metagenomes, and it could also be used for studies including other omics data (e.g., transcriptomics and proteomics). The R code of the protocol is available at https://bitbucket.org/hellekp/clinical-micro-meta-integration (accessed on 13 January 2022).

There are many multi-omics integration studies based on correlation analysis. In 2016, a multi-omics study by Kieffer et al. investigated the effect of a high-fat diet supplemented with resistant starch and found that the liver levels of the TCA metabolites fumarate and malate were decreased when mice were fed diets supplemented with resistant starch [123]. In 2019, a multi-omics integration revealed Parkinson’s disease-specific patterns in microbial-host sulfur co-metabolism that may contribute to PD severity [124]. Multi-omics integration analysis based on an order statistic algorithm was also applied to Alzheimer’s disease in 2020 [125]. In this context, a multi-omics integration analysis of metabolomics with those from other omics will help to understand the disease mechanisms and further screen key molecular markers, and then help to indicate subsequent validation experiments.

## 6. Conclusions and Prospects

A large amount of informative data are generated by the rapidly evolving field of metabolomics. At the same time, these data need to be integrated and analyzed with other omic data to be fully interpreted. The most common approach today is to simultaneously monitor transcript, protein and metabolite levels and obtain structural and dynamic changes in the underlying biological network of interest through integration analysis. This invokes the need for suitable statistical and computational methods to analyze and integrate these diverse and large amounts of data, and to visualize and map the data metabolites. Such multi-omics integration analysis can greatly contribute to the rapid identification of relevant metabolites and the biological processes when they are involved under specific research conditions.

To date, there are many tools available for processing and analyzing metabolomics data, and we reviewed and compared several of the commonly used tools. Overall, despite the undeniable validity of the tools reviewed, there are still several challenges in the field of metabolomics that need to be addressed. One of the biggest challenges is the reliable identification of known compounds, as well as the identification of unknown compounds. The fine structure of isotopes enables the determination of molecular formulae of unknowns to discover substances present in the “dark metabolome”. To achieve reliable identification of compounds, database-based search methods are often used, where retention times, accurate masses, isotopic properties, and fragment mass spectra must be provided to reliably resolve compounds in complex samples. Other challenges are mainly in the field of data integration, to support a thorough comprehensive evaluation of the experimental data and a deeper understanding of the biological processes.

With the increasing availability of multiple types of histological data, how to effectively use them to understand the observed abnormal biological mechanisms in metabolomics is still an open issue in the analysis. To achieve this goal, further development and improvement of computational techniques to identify and accurately quantify metabolites by integrating a priori knowledge and to integrate and finely visualize multi-omics data pathways is essential and will be the focus of the future field of bioinformatics.

## Figures and Tables

**Figure 1 metabolites-12-00357-f001:**
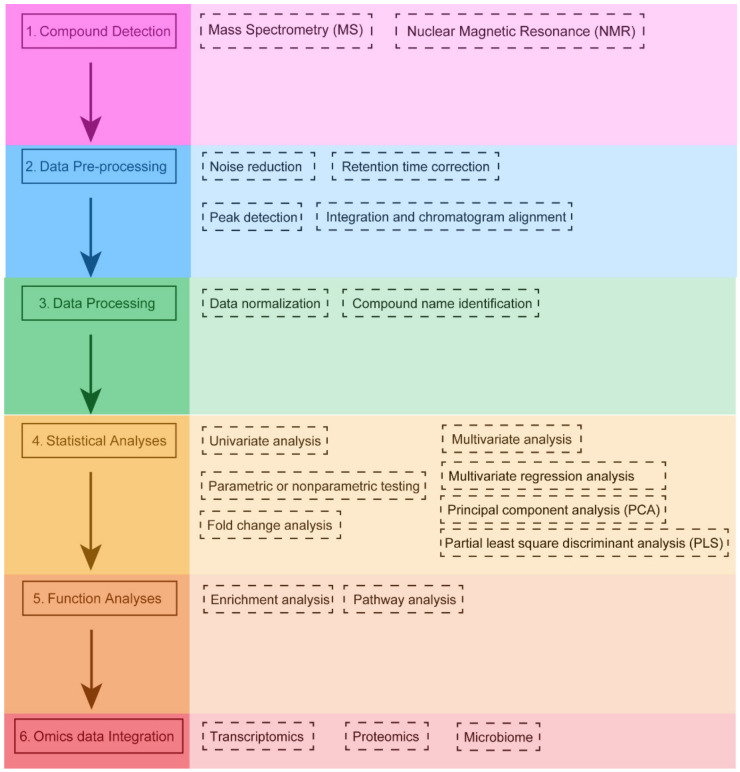
**Typical workflow of metabolomics analysis.** Metabolites are detected by using specific detection techniques (compound detection). Raw signals are then pre-processed to produce data in a suitable format for subsequent statistical analysis (data pre-processing). Then, data normalization is used to reduce the system and technical bias. For untargeted studies, metabolites are identified from spectral information in some given database (data processing). Univariate and multivariate statistical analyses are used to identify significantly expressed metabolites (statistical analyses). Next, the significantly expressed metabolites are subsequently linked to the biological context by using enrichment and pathway analysis (function analyses). Finally, metabolomics data may be integrated with other omics data (transcriptomics, proteomics, or the microbiome) to gain a comprehensive understanding of the molecular mechanisms of pathophysiological processes (Omics data Integration).

**Figure 2 metabolites-12-00357-f002:**
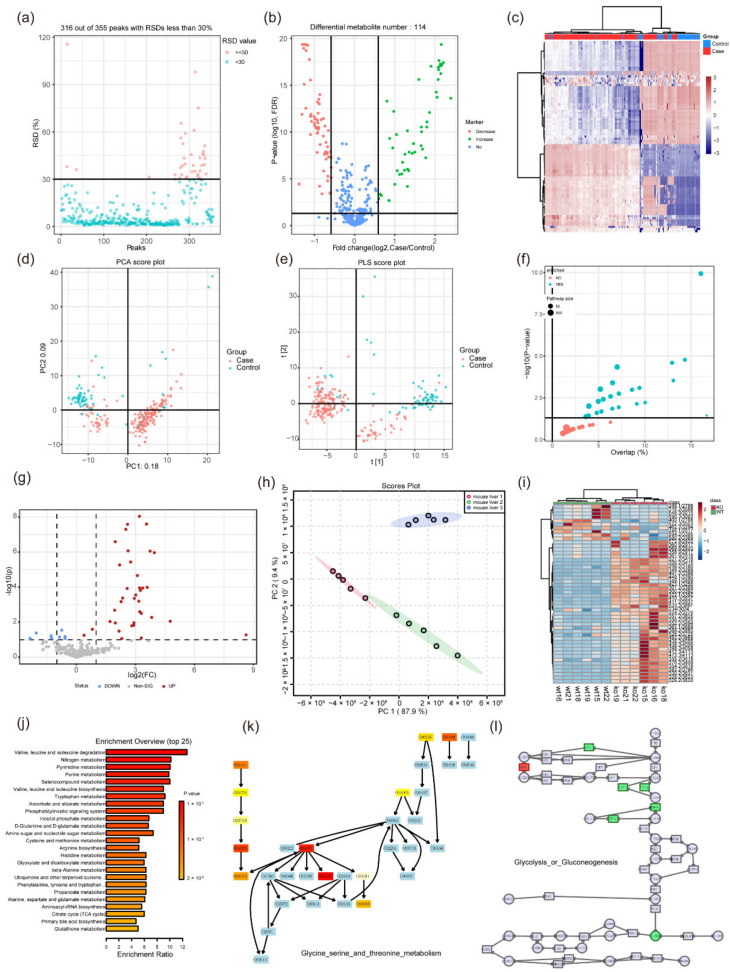
**Some graphical visualization features of MetFlow and MetaboAnalyst 5.0.** (**a**) RSD (relative standard deviation) plot in the data processing function of MetFlow. Features with a high percent RSD should be removed from the subsequent analysis (the suggested threshold is 20% for LC-MS and 30% for GC-MS). (**b**) Volcano plot and (**c**) heatmap of the differential metabolites in the statistical analysis function of MetFlow, the thresholds can be set autonomously by the submitter. (**d**) PCA analysis and (**e**) PLS analysis in MetFlow. (**f**) Pathway enrichment overview in MetFlow, each circle represents a different pathway. Circle size and color are based on the pathway size and *p*-value. (**g**) Volcano plot of the differential analysis in MetaboAnalyst 5.0. (**h**) PCA analysis plot in MetaboAnalyst 5.0. (**i**) Heatmap shows the differential metabolites in the statistical analysis function of MetaboAnalyst 5.0. (**j**) Pathway enrichment overview in MetaboAnalyst 5.0. Color shade is based on the *p*-value. (**k**) The demo-enriched metabolism pathway in MetaboAnalyst 5.0. Light blue indicates that it is not an uploaded metabolite, but instead was used as background for enrichment analysis. Red indicates the metabolite is in the uploaded data and represents the different level. (**l**) An example of joint pathway analysis in MetaboAnalyst 5.0. By uploading candidate genes and metabolites, the corresponding pathway view is generated. Squares represent genes and circles represent metabolites. Red and green indicate the different levels. All images were obtained using the example data provided by the software.

**Figure 3 metabolites-12-00357-f003:**
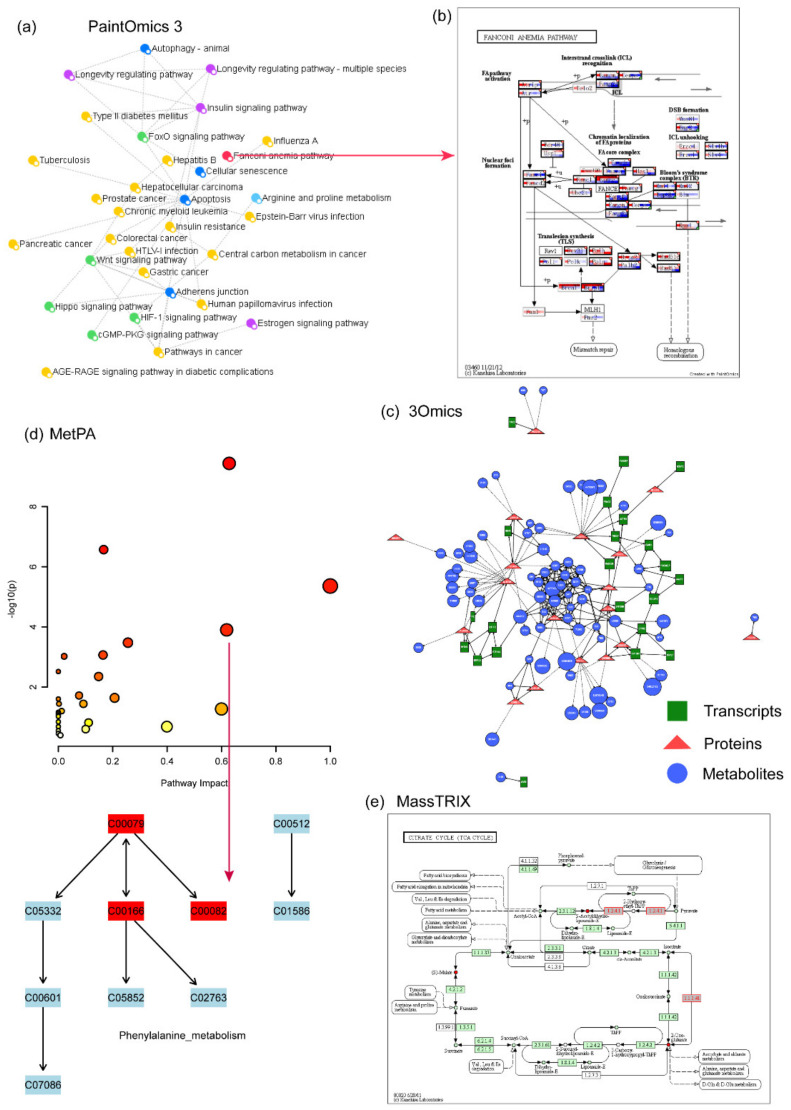
**Example of other available metabolomics data analysis tools.** (**a**) Pathway overview created by PaintOmics 3. By clicking on a circle, (**b**) the corresponding pathway view is generated, showing all genes involved in that pathway and their interactions. (**c**) A correlation network created by 3Omics. (**d**) Pathway analysis of MetPA. MetPA is now integrated into the MetaboAnalyst 5.0 platform. (**e**) Pathway analysis of MassTRIX.

**Table 1 metabolites-12-00357-t001:** Features of several most used metabolomics data analysis tools.

Name	Year	Description	Functions							
			Data Pre-Processing	Data Processing		Statistical Analyses	Pathway Enrichment Analysis	Omics Data Integration		
				Normalization	Compound Name Identification			Transcriptomics	Proteomics	Microbiome
Mzmine3	2022	MZmine3 builds on the success of MZmine 2 with many features focused on improving the user-friendly graphical	Y	Y	Y	Y	-	-	-	-
MetaboAnalyst 5.0	2021	Comprehensive web-based tool for comprehensive metabolomics data analysis, interpretation, and integration with other omics data.	Y	Y	Y	Y	Y	Y	-	-
LipidSig	2021	Web-based tool for lipidomic data analysis	Y	Y	Y	Y	-	-	-	-
MS-DIAL 4.0	2020	Lipidome atlas in MS-DIAL 4.0	Y	Y	Y	Y	-	-	-	-
El-MAVEN	2019	Fast, Robust, and User-Friendly Mass Spectrometry Data Processing Engine for Metabolomics	Y	Y	Y	-	-	-	-	-
MetFlow	2019	Interactive and integrated web server for metabolomics data cleaning and differential metabolite discovery.	Y	Y	Y	Y	Y	-	-	-
LION	2019	Web-based ontology enrichment tool for lipidomic data analysis.	-	Y	Y	Y	Y	-	-	-
Omicsnet	2018	Web-based tool for creation and visual analysis of biological networks in 3D space	-	-	-	Y	Y	Y	Y	Y
METLIN	2018	Technology platform for the identification of known and unknown metabolites and other chemical entities.	-	-	Y	-	-	-	-	-
PaintOmics 3	2018	Web-based resource for the integrated visualization of multiple omics data types onto KEGG pathway diagrams.	-	-	-	-	Y	Y	Y	-
LipiDex	2018	Integrated Software Package for High-Confidence Lipid Identification	Y	-	Y	-	-	-	-	-
LipidMatch	2017	Automated workflow for rule-based lipid identification using untargeted high-resolution tandem mass spectrometry data	Y	-	Y	-	-	-	-	-
3Omics	2013	One-click web tool for fast analysis and visualization of multi-omics data.	Y	Y	-	Y	Y	Y	Y	-
IMPaLa	2011	Pathway analysis of transcriptomics or proteomics and metabolomics data.	-	-	-	-	Y	Y	Y	-
MetPA	2010	Pathway analysis for metabolomics data.	Y	-	-	-	Y	-	-	-
MassTRIX	2008	Tool for high precision MS data annotation.	Y	-	Y	-	Y	-	-	-
MetaCore^TM^	2004	Commercial tool for functional analysis and integrated analysis of multi-omics data.	Y	-	-	-	Y	Y	Y	-

## Data Availability

Not applicable.

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
