# Peer review of "Guide to Metabolomics Analysis: A Bioinformatics Workflow"

_metabolites, 2022, doi:10.3390/metabo12040357_

Round 1

Reviewer 1 Report

A Bioinformatics Guide to Metabolomics Analysis

This review provided essential information regarding metabolomics workflow and the currently used software in metabolomics analysis. This review could bring useful information to the reader who initially do metabolomics research. However, in my opinion, the content is a bit shallow, and the quality needs to be improved to provide more comprehensive knowledge for the reader. If the manuscript contents can be improved it would be worth publishing. Please refer to the following information for more details

  • The title seems to be inappropriate with the review contents. The review aimed to provide metabolomics analysis workflow. However, the title seems to emphasize the biological interpretation part, not the whole metabolomics process.
  • Metabolomics sometimes included lipidomics, a subfield of metabolomics but has different biological interpretation procedures. According to the introduction (lines 32-34), the manuscript means metabolomics here include lipidomics. However, the whole content of this review did not mention lipidomics, which has a different identification rule and biological analysis. Please clarify this clearly in the review that the study aims to focus on metabolites, not lipids, or add lipidomics part to the review contents.
  • The summarization of abnormal metabolic pathways needs to be corrected. Firstly, please mention the sample types to where the conclusion was withdrawn as metabolomics alteration can come from bio-fluids or tumor cells samples. Second, there is no specific search term, so it is hard to say if these pathways are frequent alteration by just citing several papers. Finally, the lipid pathway was also mentioned here but was missed at the rest of the review. I would suggest to revised this part by emphasizing the importance of metabolomics in different aspects such as clinical, biomarker discovery, hypothesis generation, etc.
  • At section 2 “the analysis workflow of metabolomics”, please explain more about the differences in usage of GC-MS, LC-MS, and NMR. For example, which platform is suitable for which metabolite class. In addition, column types could be also helpful to increase the coverable compounds. In lines 121-124, the platform selected could be based on targeted compounds' chemical and physical characteristics or hypotheses, not just by the sample type.
  • When running metabolomics, quality control or internal standard is important to assure the data quality and reproducibility. Please discuss it in section 2. In addition, applying public databases could help for compound identification, but it is a second choice when the in-house library is not available. Therefore, compound ID by authentic standard should be mentioned before the published databases.
  • In section 3 “statistical analysis in metabolomics”, the multivariate analysis is not only included Anova, PCA, PLS-DA. The selection of multivariate analysis should also refer to the aim of the analysis. For instance, PCA would provide information about data structure and outlier information. In addition, other multivariate analysis should be mentioned, like cluster analysis, KNN, and machine learning for generation model can be applied in metabolomics. Please discuss it more.
  • Section 4 “software tools” is the most helpful information provided in the review. Therefore, I would suggest adding more software that is missed in the current review. First of all, since this is a review of metabolomics analysis workflow, it should discuss about pre-processing data software too. Several favored software includes MS-DIAL, Mzmine3, El-MAVEN. Each of these software has pros and cons. For example, MS-DIAL and Mzmine can process different platforms like GS-MS, LC-MS, NMR. For MS-DIAL, it provides multiple data-acquisition processing and includes the spectral for compound ID. It also includes normalization and statistical analysis options. Please discuss that software too as they are popular.
  • Regarding lipidomics, software such as Lipidmatch and Lipidex are worth mentioning
  • For multi-omics integration, Omicsnet, a web-based allows generating a gene-metabolite pathway interaction, should be mentioned as they are useful to hypothesis generation by just inputting the list of gene and metabolite, which can easily get from published papers. For lipidomics, LION software, Lipidsig should be mentioned.
  • Last but not least, please discuss the challenge of metabolomics analysis and the standardization recommendations to provide a reproducible result for future research.

In general, it is an essential review. However, more detailed information should be added to provide comprehensive and helpful information to the reader.

Reviewer 2 Report

This is a very interesting manuscript, however the deviation from the proposed format is making it hard to follow and fully understand the intention of the authors. I would suggest that the authors may consider the following:

Introduction: An essential step of the introduction would be to also present relevant work done in the respected field of research as well as clarify how this present work adds up to what is already available in the literature and what is the gap that this Bioinformatics Guide to Metabolomics Analysis is here to help with. Perhaps the authors could consider including such information here.

Lines 65-100: As per the title and abstract, it is clear that this is not the main scope of this review. Kindly consider elaborating as to what is the use of this part and also note that it is essential for the reader to have clear information regarding the studies presented in this segment of the manuscript. On that note, perhaps the authors would like to consider including a brief description regarding the process that led to the selection of these studies.

Also, at this point (lines 65-100) the manuscript seems to be moving further on to the results’ segment and therefore it may be beneficial for the reader to be guided as such by a heading. Understandably, it may be difficult to cut this review into the segments required by the journal (introduction, results, discussion, materials and methods, and finally conclusions) but it is essential to keep the format relevant to the journal’s guidelines and also provide the reader with a familiar and gradually developing manuscript setting. On that note, I invite the authors to consider revisiting the text and try to reorganize it as per the journal’s guidelines.

Line 113-114: Kindly consider removing the text “This section may be divided by subheadings. It should provide a concise and precise description of the experimental results, their interpretation, as well as the experimental conclusions that can be drawn.” as it appears to be the instruction left from the initial template.

Materials and methods: As stated before, this segment seems to be missing from the manuscript and it is an essential part considering that the information regarding the steps taken to construct this manuscript is not reported. In this part, it would be crucial to present the evaluations made by the authors in order to include or exclude the software tools of metabolomics described.  For example, were the most widely used tools selected for inclusion in the review? And if so, which were the selection criteria?  (Perhaps, the number of citations on Web of Science or the use of the tool being reported by Metabolomics Society). The authors have briefly reported the scope of the manuscript in lines 103-105 however, I would invite you to kindly consider including these steps in the text lines 209-215.

Line 124: Perhaps the authors would consider including the text of lines 39-64 here, as it further explains the strengths and limitations of each technique which is probably more relevant with this part of the manuscript.

Line 139-142: Kindly consider elaborating as to why the tools mentioned here are not relevant for the scope of this review and therefore are not presented in further detail.

Line 143: Since the authors have decided to present this segment in steps (which may be very beneficial for the reader) kindly consider elaborating if here starts the third step.

Line 28-376: It would seem that the introduction, materials and methods, and results are somewhat interlinked within this part of the manuscript. I strongly recommend that the authors revisit this great work they made and just put it in a format that would be more beneficial for the reader and also meet the journal’s guidelines.

Line 380: As it reads, in this part of the manuscript the reader would expect to see the discussion part. Here of course the authors have made a discussion about their overall observations as well as previous findings and utilization of the integration algorithm of multi-omics data, however, the initial findings presented (metabolic pathways in 7 malignant tumors and 3 metabolism-related diseases) are not reported anywhere. Please consider explaining the purpose of these findings for the scope of this review, otherwise, there is no link to it in the text (or even the title of the review) and the authors may remove it.

The discussion part of the manuscript must be revised in order to meet the scope and clarify the findings.

Line 421-422: Kindly consider removing the text “This section is not mandatory but can be added to the manuscript if the discussion is unusually long or complex” as it appears to be the instruction left from the initial template.

Overall, the authors have some great material to work with and made a good effort to provide relevant information. Unfortunately, the management of the text does not do justice to their work and needs to be revisited.

Round 2

Reviewer 1 Report

The authors have mostly solved all my concerns and provided an up-to-date software in metabolomics and lipidomics.

However, there are few minor points that should be revisited in the new manuscript version. Please refer to the following information.

Line 153-155: Please mention about the “reporting standards for metabolite annotation and identification” (https://www.ncbi.nlm.nih.gov/pmc/articles/PMC3853013/)

Line 226-235: One of a common mistake when using PLS-DA is the overfitting problem. Therefore, cross-validation should be applied when using PLS-DA. In addition, the result of PLS-DA should be interpreted along with PCA to avoid the overfitting problem.  Please emphasize this point in the current review as it would help research paying attention when using PLS-DA.

Line 487-488: “MS-DIAL, Mzmine3 and EI-MAVEN” can help from the data pre-procession, normalization and identification, not just the normalization.

Overall, the quality of review has significantly improved. I believe it would bring valuable information to future metabolomics research.

Reviewer 2 Report

The authors have made all recommended changes to the manuscript, which is now very much improved.

I invite the authors to kindly revisit the newly added text as there are minor grammatical issues in some sentences.

A minor suggestion for rephrasing (clearly optional) Line 77: "However, these reviews have not described the overall metabolomics analysis process" Maybe this is not very nicely put for the fellow researchers, perhaps something along the lines of: "A detailed guidance for the analysis process would be an essential addition to this previous work" would be an option the authors would kindly consider.
